# Role of RNA Alternative Splicing in T Cell Function and Disease

**DOI:** 10.3390/genes14101896

**Published:** 2023-09-30

**Authors:** Shefali Banerjee, Gaddiel Galarza-Muñoz, Mariano A. Garcia-Blanco

**Affiliations:** 1Department of Microbiology, Immunology and Cancer Biology, University of Virginia, Charlottesville, VA 22903, USA; marianogb@virginia.edu; 2Department of Biochemistry and Molecular Biology, University of Texas Medical Branch, Galveston, TX 77550, USA; 3Autoimmmunity Biologic Solutions, Inc., K2bio, Houston, TX 77550, USA; gaddiel@abstherapeutics.com

**Keywords:** alternative splicing, T cells, immunity, autoimmune disease

## Abstract

Alternative RNA splicing, a ubiquitous mechanism of gene regulation in eukaryotes, expands genome coding capacity and proteomic diversity. It has essential roles in all aspects of human physiology, including immunity. This review highlights the importance of RNA alternative splicing in regulating immune T cell function. We discuss how mutations that affect the alternative splicing of T cell factors can contribute to abnormal T cell function and ultimately lead to autoimmune diseases. We also explore the potential applications of strategies that target the alternative splicing changes of T cell factors. These strategies could help design therapeutic approaches to treat autoimmune disorders and improve immunotherapy.

## 1. Introduction

T cells are crucial in establishing immune responses, maintaining immunological memory, and regulating immune homeostasis. T cells are first formed in the bone marrow and then move to the thymus for maturation and selection before subsequently being exported to the periphery. During the thymic stages of development, T cells undergo lineage commitment to become either CD4^+^ or CD8^+^ T cells [1]. CD4^+^ T cells bind to MHC class II molecules, while CD8^+^ T cells bind to MHC class I molecules, which are found on antigen-presenting cells (APCs). Mature and immunologically naïve T cells express cell surface receptors (TCR) that recognize and bind to the short peptide antigens associated with MHC molecules and, in the presence of appropriate co-stimulatory signals, induce a downstream signaling cascade, resulting in T cell activation. When T cells are activated, they undergo clonal expansion and differentiate into CD4^+^ helper (Th1, Th2, and Th17) or CD8^+^ cytotoxic effector cells. These cells produce cytokines to respond to immune challenges effectively. Most effector T cells are short-lived, while a subset develops into memory T cells (CD4^+^ and CD8^+^) to maintain long-term immunity [2]. Another subtype of mature T cells includes the T regulatory cells (T regs) responsible for suppressing autoimmunity and excessive proinflammatory responses to maintain self-tolerance and immune homeostasis [3]. T cells are remarkably capable of responding and adapting to changing environmental conditions. This ability to adapt is achieved by precisely regulating complex gene and protein expression networks. Alternative RNA splicing is essential in regulating gene expression in T cell physiology.

RNA splicing plays a crucial role in the eukaryotic RNA metabolism and can be classified into constitutive and alternative splicing. Constitutive splicing refers to the removal of introns and the joining of consecutive exons, while alternative splicing refers to the variable inclusion of exons and introns. The spliceosome machinery catalyzes the splicing reaction by assembling on an intron after recognizing and binding to the 5′ splice site (5′SS) and 3′ splice site (3′SS) sequences that flank the intron. Most of the introns are spliced by the major spliceosome, which consists of U1, U2, U4/U6, and U5 small nuclear ribonucleoproteins (snRNPs). A closely related machinery called the minor spliceosome splices a small class of introns. This review will focus on the alternative splicing changes mediated by the major spliceosome machinery. Several factors such as the strength of the 5’ and 3’ splice sites, the binding of RNA-binding proteins to regulatory sequence elements in introns or exons, and the interactions between auxiliary splicing factors and the core spliceosome machinery can dictate whether an exon or intron is alternatively spliced. The regulatory sequence elements that activate splicing are classified as exonic (ESE) and intronic splicing enhancers (ISE), and the sequence elements that inhibit splicing are classified as exonic (ESS) or intronic splicing silencers (ISS). RNA alternative splicing modifies the sequence of a transcript, and depending on whether these changes occur in the coding or non-coding regulatory regions of the transcript, it can impact mRNA stability, transport, protein expression, and function. Alternative splicing is a widespread mechanism for regulating protein expression and function and plays an important role in shaping the adaptive immune response. Many immune-related genes, which include cell surface receptors (CD3, CD28, CTLA-4, IL7R), kinases and phosphatases (MAP4K2, MAP3K7, MAP2K7, CD45), transcription factors (LEF1, GATA3, FOXP3), and RNA-binding proteins (CELF2, TIA-1), undergo alternative splicing during an immune response. Some of these alternative splicing changes are unique to T cells [4]. For example, both naïve B and T cells express the full-length isoform of the transmembrane protein tyrosine phosphatase CD45. In contrast, only activated and memory T cells express the shorter isoforms that arise from the regulated alternative splicing of exons 4, 5, and 6 in the CD45 pre-mRNA [5]. CD45 initiates TCR signaling by dephosphorylating the inhibitory tyrosine residues in Src-kinases. T cells expressing the shorter isoforms of CD45 have reduced phosphatase activity. The alternative splicing of CD45 pre-mRNA in activated T cells acts as a feedback mechanism to attenuate prolonged TCR signaling [5].

Apart from CD45, several cases of gene-specific and overall alternative splicing alterations play a crucial role in different aspects of T cell functionality, such as differentiation, activation, apoptotic signaling, and T cell proliferation [6,7,8,9]. As a result, it is not unexpected that irregular splicing changes could lead to abnormal T cell functions and immune-related disorders. It is worth noting that some alternatively spliced variants of T cell factors, such as IL2RA, IL7R, CD45, CD44, and CTLA-4, are associated with an increased risk of developing autoimmune diseases, such as multiple sclerosis, type 1 diabetes, and rheumatoid arthritis [10,11,12,13].

This review focuses on how changes in the alternative splicing of T cell factors can impact T cell function and contribute to autoimmune diseases. To achieve this, we utilized two databases: ClinVar, which aggregates information about genomic variation and its relationship to human health, and PubMed. We searched the ClinVar database for mutations associated with autoimmunity and T cell factors. Our focus was on splice site mutations and splice variants identified as pathogenic, indicating that these mutations have been found in individuals with immune disorders. The complete list of splice site mutations in different T cell factors is listed in Appendix A. Next, we searched for studies on RNA alternative splicing and T cells in PubMed. We focused on studies that identified alternatively spliced genes associated with the pathogenic splice site mutations we identified in our ClinVar search. This review provides examples of how changes in the alternative splicing of T cell factors such as receptors, kinases, transcription factors, cytokines, and RNA-binding proteins result in abnormal T cell function and immune disorders.

## 2. Cell Surface Receptors

Our search identified several pathogenic splice site mutations in T cell surface receptors such as CD3, CD81, CTLA-4, Fas, ICOS, and IL2R (Table 1). Here, we elaborate on the role of alternative splicing and the potential implications of splice site mutations in Fas and CTLA-4 on T cell function.

### 2.1. Fas (CD95/Apo-1)

Fas, which is a member of the tumor necrosis receptor (TNFR) family, is expressed in lymphoid, myeloid, and non-hematopoietic cell types, and its ligand, FasL—a member of the TNF family—is only expressed in CD8^+^ or activated CD4^+^ T cells. FAS–FASL binding induces apoptotic cell death through the caspase signaling pathway and is essential for maintaining immune homeostasis. FAS-mediated apoptotic signaling is required to remove autoreactive T cells in the thymus and eliminate T cells that are activated without appropriate costimulatory signals [14]. The inhibition of Fas/FasL signaling results in uncontrolled lymphocyte proliferation, which can lead to autoimmune diseases such as multiple sclerosis. Multiple sclerosis is a T-cell-mediated inflammatory disease, where the body’s immune system attacks its central nervous system (CNS), and is characterized by demyelination and axonal loss. Notably, most patients with autoimmune lymphoproliferative syndrome (ALPS) exhibit mutations in the Fas/FasL genes [15]. ALPS is a rare genetic disorder associated with the uncontrolled proliferation of lymphocytes, which can result in debilitating health conditions such as autoimmune diseases and lymphomas.

Fas has two transcript isoforms: a full-length membrane-bound isoform and a soluble protein (sFas) isoform lacking exon six, which encodes for the transmembrane domain. sFAS competitively binds to FasL and inhibits its interaction with the Fas receptor on T cells, inhibiting downstream apoptotic signaling [16,17] (Figure 1). Several RNA-binding proteins such as PTB, HuR, hnRNPC, SRSF6, and RBM5 induce Fas exon 6 skipping [18,19,20,21,22]. One model proposes that PTB binds to an exonic splicing silencer (ESS) in exon 6 and inhibits the binding of U2AF to the 3′SS of the upstream intron 5, thus disrupting exon definition and promoting exon skipping [22]. Another model suggests that both HuR and hnRNPC bind to the ESS in exon 6 and cooperatively inhibit exon definition by preventing the recruitment of U1snRNP to the 5′SS of intron 6 as well as U2AF recruitment to the 3′SS of intron 5 [19]. Contrarily, TIA-1/TIAR recruits U1snRNP to the 5′SS of downstream intron 6 and promotes exon 6 definition and inclusion [21]. Apart from RNA-binding proteins regulating Fas exon 6 splicing, a long non-coding RNA (lncRNA) termed Fas antisense 1 lncRNA or Saf also promotes exon 6 skipping by interacting with Fas pre-mRNA and splicing factor 45 [23].

Several variants of exon 6-skipped transcript encoding for different soluble Fas isoforms have been identified in patients with autoimmune lymphoproliferative syndrome [24]. The Clinvar database lists 15 pathogenic splice site mutations in the *Fas* gene found in patients with autoimmune lymphoproliferative syndrome (Table 1). One of the listed splice site mutations suggested to promote the skipping of exon 6 in *Fas* pre-mRNA is a G>C (c.506-1G>C) change in the 3′SS of intron 5 [25]. The other splice site mutations are in introns 3, 4, 6, and 7. Splice site mutations that change the 5′SS and 3′SS of intron 7 (c.651+1G>A, and c.652-1G>A) are predicted to disrupt RNA splicing and result in absent or disrupted Fas protein. Another splice site mutation in 5′SS of intron 8 (c.676+1G>A) is predicted to encode for a splice isoform lacking exon 8 and a disrupted C-terminal region. The aberrant alternative splicing of *Fas* primary transcripts can impair Fas-mediated apoptotic signaling in T cells, which can contribute to the uncontrolled proliferation of activated T cells, as seen in cutaneous T-cell lymphomas (CTCLs), a group of malignancies derived from skin-homing T cells [16].

### 2.2. Cytotoxic T lymphocyte Antigen 4 (CTLA-4)

CTLA-4 is a member of the immunoglobulin receptor subfamily and is predominantly expressed in CD4^+^ T effector cells and T reg cells. CTLA-4 is homologous to CD28 and can competitively bind to its ligands, CD80/CD86, and inhibit the CD28-mediated costimulatory signaling in activated T cells. CTLA-4 functions as an immunosuppressor by blocking the expression of IL-2 and stopping the cell cycle progression from the G1 to the S phase of activated T cells [26]. CTLA-4 is alternatively spliced to produce two protein isoforms: a membrane-bound receptor protein and a soluble protein isoform (sCTLA-4) lacking the transmembrane domain. These two CTLA-4 isoforms arise from the alternative splicing of exon 3, which encodes the transmembrane domain [27]. sCTLA-4 also competitively binds to CD28 ligands and, like CTLA-4, has an immunosuppressive function [13]. In resting naïve T cells, sCTLA-4 is the most abundant isoform, whereas activated T cells predominantly express the membrane-bound CTLA-4 isoform [26]. Abnormal CTLA-4 isoform ratios have been associated with immune disorders such as Grave’s disease [27], a T-cell-mediated autoimmune disorder that results in hyperthyroidism. The single nucleotide polymorphisms in the *CTLA4* gene are associated with several autoimmune disorders such as rheumatoid arthritis, type I diabetes, and systemic lupus erythematosus (SLE) [10,13,27,28,29,30]. Two pathogenic splice site mutations in CTLA-4 are present in the ClinVar database, and both these mutations (c.458-1G>C and c.458-1G>T) disrupt the 3′SS in intron 2 (Table 1). The functional consequences of these transcript variants are not characterized but are predicted to either encode for transcripts directed for nonsense-mediated decay or loss-of-function isoforms. CTLA-4 has immunoregulatory roles in activated T cells and T-reg cells, and splice site variants that can alter CTLA-4 alternative splicing outcomes can result in T cell hyperactivation, immunodeficiency, and a variable degree of immune dysregulation [31]. 

## 3. Intracellular Signaling Factors

This review section will focus on how alternative splicing modifications affect cellular factors, such as intracellular kinases. These factors mediate signals downstream of the TCR complex and regulate various signal transduction pathways.

### 3.1. Lymphocyte-Specific Tyrosine Kinase (Lck)

Lck is a Src family of protein tyrosine kinase constitutively expressed in CD4^+^ and CD8^+^ T cells. When attached to the membrane, Lck interacts with the CD4 or CD8 co-receptors bound to MHC molecules and triggers a downstream signaling cascade by phosphorylating the immunoreceptor tyrosine-based activation motif (ITAM) within the TCR complex and ZAP70 (CD3 zeta chain associated 70kDa protein) [32]. Lck is critical for the early propagation and modulation of TCR signaling, and a loss of Lck activity is associated with impaired T cell development, differentiation, and effector functions [33,34,35]. Decreased Lck activity contributes to type I diabetes pathogenesis [36]. Patients with Lck deficiency frequently present with immune dysregulation and autoimmunity [37]. 

*Lck* encodes for a full-length transcript isoform and an exon 7 skipped isoform, which lacks the ATP-binding domain required for its kinase activity. This catalytically inactive Lck isoform accounts for almost 15% of all Lck transcript isoforms in T cells. It is believed to have a regulatory role by competitively binding to Lck targets and inhibiting their phosphorylation [38]. The splice site mutations in the *Lck* gene that promote the skipping of exon 7 are associated with abnormal T cell function. Another rare alternatively spliced isoform, which results in the retention of intron 2 and the skipping of exon 7, was observed in T cells from type 1 diabetic patients [39]. The retained intron does not disrupt the reading frame and instead results in the insertion of 58 amino acids upstream of the receptor-interacting domain. This isoform’s functional consequence is unclear, but it is predicted to have failed interactions with the TCR receptor and downstream targets. Another splice isoform that arises from the alternative splicing of exon 1 has a modified N-terminus, which lacks the regions required for membrane association. Varying ratios of the full-length and exon 1 skipped transcript isoforms are observed in patients with T-cell acute lymphoblastic leukemia (T-ALL) [40]. The ClinVar database has a splice site mutation that disrupts the 5′SS (c.481+2T>G) of intron 6 in Lck pre-mRNA (Table 1). Although the functional significance of this splice isoform is unclear, the disruption of the 5′SS sequence could hinder the binding of U1snRNP and spliceosome assembly on intron 6. This could result in intron retention or exon skipping, leading to transcript isoforms with premature stop codons that may undergo nonsense-mediated decay. Another splice site mutation that changes the 3′SS (A>G) of intron 2 in Lck pre-mRNA is also associated with abnormal T cell function. The resulting splice isoform lacks exon 3, which introduces a premature stop codon. This splice isoform is linked to a Lck deficiency that abrogates the initiation of TCR signaling. Such genetic polymorphisms, resulting in splice variants that disrupt Lck-mediated TCR signaling and T cell function, can increase susceptibility to autoimmune and immunodeficiency diseases [41,42]. 

### 3.2. CD3 Zeta Chain Associated 70kDa Protein (ZAP70)

ZAP70, which is a protein tyrosine kinase involved in the initiation of TCR signaling, undergoes activation upon binding to phosphorylated ITAMs in CD3 receptor complex and subsequent phosphorylation by Lck [43]. Activated ZAP70 phosphorylates downstream adapter proteins, acting as the linker for the activation of T cells (LAT) and the SH2-domain-containing leukocyte protein of 76 kDa (SLP-76), which function as scaffolds to recruit additional signaling molecules and mediate signal transduction leading to T cell activation, differentiation, and proliferation [43]. 

Two alternatively spliced isoforms of ZAP70 are expressed in T cells. The full-length isoform encodes for the full-length ZAP70 protein containing two N-terminal SH2-domains, which bind to the CD3 receptor, and a C-terminal kinase domain connected by the interdomain B. The shorter isoform encodes for a catalytically active truncated protein lacking receptor-binding domains and a region of interdomain B [44]. Unlike its full-length isoform, the truncated ZAP70 isoform is not recruited to the immune synapse and fails to transduce the TCR-mediating signaling [44]. Another ZAP70 splice isoform, which introduces a new 3′SS in intron 7, was identified in a patient with severe combined immunodeficiency syndrome. The resulting transcript isoform, the major isoform expressed in the patient, contains a premature stop codon and undergoes non-sense-mediated decay [45]. Additional splice site mutations that introduce a premature stop codon in ZAP70 transcripts are also listed in the ClinVar database (Table 1). One of these variants includes a splice site mutation that affects the 3′SS (c.703-1G>A) in intron 5 and is present in patients diagnosed with Epstein–Barr virus-associated lymphoproliferative disorder/lymphoma [46]. The disruption of this splice site is predicted to introduce a premature codon, and the resulting transcript isoform is expected to undergo nonsense-mediated decay. Genetic variants that produce aberrant ZAP-70 isoforms can affect T cell development and homeostasis by interfering with TCR signaling initiation.

## 4. Transcription Factors

Transcription factors induce the expression of immune effector proteins following interactions with upstream signaling molecules. In this section, we focus on the example of FOXP3 and elaborate on how alternative splicing regulates its function. 

### Forkhead Transcription Factor 3 (FOXP3)

FOXP3 belongs to the Forkhead transcription factor family and plays a critical role in the development, maintenance, and function of T regulatory cells [3,47,48,49] The FOXP3 protein comprises an N-terminal proline-rich repressor domain, the zinc finger and leucine zipper domains necessary for homo and heterodimerization, and a C-terminal Forkhead DNA-binding domain. Alternative splicing plays a vital role in regulating FOXP3 expression and function. Our group has shown that DDX39B (a DEAD-box RNA helicase) regulates the efficient removal of several *FOXP3* introns. A lack of DDX39B activity decreases FOXP3 RNA and protein expression due to increased FOXP3 intron retention [50]. 

The alternative splicing of *FOXP3* pre-mRNA results in the expression of transcript isoforms that vary in their domain composition. It must be noted that the alternative splicing of *FOXP3* pre-mRNA is species-specific and unique to humans, so it is not present in mice. Three FOXP3 transcript isoforms are produced by alternative splicing: full-length, exon 2 skipped (FOXP3del2), and exon 2 and exon 7 skipped (FOXP3del2del7) [51]. In *FOXP3* mRNA, exon 2 encodes for a part of the repressor domain, and exon 7 encodes for a part of the leucine zipper domain. The full-length and exon 2 skipped FOXP3 isoforms have overlapping immunosuppressive functions. For example, they both can bind to NF-kB and inhibit NF-kB-mediated proinflammatory cytokine production [51]. The FOXP3 isoform lacking exons 2 and 7 functions as a dominant negative isoform and inhibits the activity of other FOXP3 isoforms. The exons 2 and 7 skipped FOXP3 isoform is thought to mediate the differentiation of Treg cells to Th17 cells by inhibiting FOXP3 activity [52]. An increased expression of this dominant negative isoform is observed in patients suffering from Crohn’s disease, a chronic inflammatory disease of the gastrointestinal tract. The evidence suggests that IL1*β* signaling is responsible for the increased skipping of exons 2 and 7 from the FOXP3 pre-mRNA [52].

The importance of FOXP3 in immunoregulation is highlighted by the significant associations between inactivating FOXP3 mutations and increased susceptibility to autoimmune diseases such as the IPEX (immunodysregulation, polyendocrinopathy, enteropathy, X-linked) syndrome [51]. Of all the 44 distinct FOXP3 mutations associated with IPEX syndrome, 11 were splicing-associated mutations. Most of these mutations were present within exon 7 and in adjacent introns. Though these splice variants are not functionally characterized, they may lead to exon 7 skipping. The ClinVar database lists five splice site mutations in the *FOXP3* gene observed in patients with IPEX syndrome. Three of these mutations are found to affect the 5′SS of intron 2 (c.210+1G>A, c.210+G>C, and c.210+G>T) and disrupt RNA splicing. The resulting transcript isoforms undergo nonsense-mediated decay or encode for disrupted FOXP3 proteins. Another rare splice variant (c.-23+1G>T), located between the untranslated exon 1 and intron 1 in the *FOXP3*, is also predicted to disrupt RNA splicing. FOXP3 expression is necessary for defining the phenotype and function of Tregs. FOXP3 splice variants, which disrupt its function, can downregulate immune tolerance.

## 5. Factors Mediating Global Alternative Splicing Changes in T Cells

The abovementioned factors are examples of gene-specific alternative splicing changes that modulate T cell function and are associated with autoimmune disorders. In the next part of the review, we will focus on factors such as RNA-binding proteins and proteins such as methyltransferases, which carry out post-translational modifications. These proteins can mediate alternative splicing changes across multiple genes in T cells. 

The availability of different transcriptome profiling methods, combined with the progress made in computational tools for analyzing large-scale RNA-sequencing data, has been crucial in studying changes in alternative RNA splicing in single cells, tissues, and organ systems. A study by Kristen Lynch and colleagues [8] has shown that upon stimulation, T cells undergo regulated alternative RNA splicing changes in several immune-related genes (CD45, MAP2K7, TRAF3, Fyn, etc.), and there is a significant difference between the repertoire of alternatively spliced isoforms in naïve and activated T cells. This study indicates that the control of alternative splicing relies on the nature of the stimuli and that distinct activating stimuli can lead to the expression of different alternatively spliced isoforms. For example, the activation of T cells with either PMA (phorbol 12-myristate 13-acetate) or ionomycin induces alternative splicing changes in different classes of genes. This might partly be mediated by the expression of RNA-binding proteins such as hnRNPL/LL. It is worth noting that the exons included more upon PMA stimulation were enriched for sequence motifs required for hnRNPL/LL binding. In another study, Blake et al. investigated whether the CD28-mediated co-stimulation of activated T cells contributes to alternative splicing changes [6]. They found that though CD28 signaling alone has minimal impact on splicing, it enhances the extent of change for TCR-induced alternative splicing events. These CD28-enhanced splicing events were observed in the apoptotic signaling pathway genes, namely caspase-9, Bax, and Bim. The increase in the expression of the specific alternatively spliced isoforms was responsible for the increased cell proliferation required for the clonal expansion of activated T cells during an immune response. 

The role of RNA-binding proteins in mediating alternative splicing changes is well described in relation to the regulated alternative splicing of Fas pre-mRNA (as explained previously). RNA-binding proteins such as hNRPL/LL, PTB, HuR, and TIA-1/TIAR regulate Fas exon 6 alternative splicing by binding to sequence motifs within the exon and adjacent introns in Fas pre-mRNA. Some of these proteins, such as hnRNPL/LL, are upregulated upon T cell activation, which might result in increased the binding of hnRNPL/LL to the exonic and intronic sequence motifs of pre-mRNAs and promote exon skipping, as seen in the example of Fas pre-mRNA splicing. 

The DEAD-box RNA helicase DDX39B is another example of an RNA-binding protein essential in regulating alternative splicing outcomes of immune-related genes such as IL7R and FOXP3. Our group has previously shown that DDX39B regulates the alternative splicing of IL7R exon 6 and promotes the inclusion of exon 6 in IL7R mRNA to produce the membrane-bound isoform [12]. According to the study, there is a genetic association between a 5′UTR SNP in the DDX39B gene that decreases its expression and a SNP in the IL7R gene that increases the risk of developing MS. DDX39B was also found to regulate the splicing of FOXP3 introns, and reduced DDX39B expression results in a decrease in FOXP3 expression in Tregs [50]. Based on these examples, DDX39B might play significant immunoregulatory roles in T cells.

Proteins that carry out post-transcriptional and post-translational modifications can also influence global changes in alternative splicing networks in T cells. Studies have shown that Protein Arginine Methyl Transferases (PRMTs) such as PRMT5 are responsible for the post-translation methylation of arginine residues on proteins and can affect alternative splicing changes in T cells [53]. Some known targets of PRMT5 include U snRNP Sm proteins (SMB, SMB’, SMD1, and SMD3) and RNA-binding proteins such as hnRNPK. A study showed that PRMT5 regulates almost 16% of the alternative splicing outcomes in activated T cells [53]. The absence of PRMT5 activity leads to poor T cell proliferation and survival due to impaired IL2 and IL7 signaling [54]. This can be partly attributed to the increased expression of the abnormally spliced transcript isoforms of the common γ-chain (γc) cytokine receptor, a common receptor for both the IL2 and IL7 cytokines, resulting in decreased γ-chain expression. 

Kinases such as CDC-like kinases (CLKs), serine arginine protein kinases (SRPKs), pre-mRNA processing factor 4 kinase (PRP4K), etc., are another class of proteins that carry out post-translational modifications of splicing factors [55]. The phosphorylation of spliceosome components such as SRSF1 and PRP28, mediated by SRPK1 and SRPK2, respectively, is required for spliceosome assembly [56,57]. The splicing factor SRSF1 regulates the transcriptional program of Treg cells, and mice with aberrant SRSF1 activity develop autoimmune disorders [58,59]. Alterations in the phosphorylation states of splicing factors can impact their nuclear localization and distribution. The CLK1-mediated phosphorylation of the splicing factor SPF45 is required for its role in promoting exon 6 skipping in Fas pre-mRNA, which has significant consequences in regulating T cell survival and proliferation [60]. SC35 is a splicing factor that regulates the alternative splicing of CD44 and CD45 genes in T cells, and its activity is dependent on the PKC-theta-mediated phosphorylation of the serine residues in its RNA-recognition motifs [61,62]. 

## 6. Therapeutic Modifications of Alternatively Spliced Variants in T Cells

It is evident that RNA splicing plays a crucial role in regulating T cell signaling responses and immune function. Hence, developing strategies to target T-cell-specific splicing events can be valuable for treating autoimmune disorders and cancers. Small molecule splicing modulators that target different components of the splicing machinery are still in the early stages of drug development. Pladienolide derivatives such as E7107 and H3B-8800, which bind to SF3B1 and prevent its conformational changes, have shown limited clinical potential [63]. SF3B1 is a core component of the U2 snRNP involved in branchpoint recognition and early spliceosome assembly. Inhibitors of CDC-like kinases (CLKs) have shown therapeutic potential in treating diseases such as Duchenne muscular dystrophy (DMD). DMD is a neuromuscular disorder that arises from a lack of dystrophin activity. CLKs phosphorylate serine/arginine proteins (SR proteins), influence their association with pre-mRNA, and modulate alternative splicing. The CLK inhibitor (TG693) inhibits the phosphorylation of SR proteins and promotes exon skipping and the production of a truncated functional dystrophin protein [64]. Another CLK inhibitor (CaNDY) has shown promise in correcting the mis-splicing of the CFTR gene in cystic fibrosis (CF). CF is an autosomal-recessive monogenic disease caused by mutations in the *CFTR* gene that decrease CFTR channel expression and activity. The most common splicing mutation in CFTR (3849+10 kb C>T deep-intronic mutation) causes the inclusion of a pseudo exon within intron 22, and CaNDY prevents the inclusion of this pseudo exon and restores CFTR channel expression and activity [65]. Another potential class of splicing modulators includes PRTM5 inhibitors such as CPM5, which has immunosuppressive activity in an experimental autoimmune encephalomyelitis mouse (EAE) model [66]. RBM39 is another splicing factor that has been extensively explored as a potential therapeutic target for modulating alternative splicing in cancers. Notably, RBM39 is also highly expressed in immune cells such as CD4^+^ and CD8^+^ T cells and could potentially modulate alternative splicing in these cells. Several small molecule inhibitors that target RBM39 for proteasomal degradation are in different stages of clinical trials for treating leukemias and lymphomas [67]. However, it must be noted that such small-molecule splice modulators can cause widespread changes in pre-mRNA splicing, leading to non-specific clinical outcomes.

A more targeted approach to modulating RNA splicing outcomes is through antisense splice-switching oligonucleotides (ASOs) [68]. ASOs bind to their complementary sequences in the pre-mRNA and can promote the inclusion or exclusion of exons. These ASOs block RNA–RNA and RNA–protein interactions between pre-mRNA and splicing factors upon binding to pre-mRNA. ASO-based therapies have shown significant clinical potential in treating diseases caused by abnormal RNA splicing. Several splice-modulating ASOs, including eteplirsen, golodirsen, and casimersen, have been approved by the FDA to treat Duchenne muscular dystrophy [69,70,71]. Another example of FDA-approved ASO-based therapy is nusinersen for treating spinal muscular atrophy (SMA) [72]. An in vitro study explored the possibility of using ASO to modulate the alternative splicing of hTERT—which is the catalytic subunit of the telomerase complex—to inhibit telomerase activity in CD4^+^ T cells [73]. The ASO binds to a region within intron 8 of hTERT pre-mRNA and prevents the interactions between the pre-mRNA and splicing regulatory proteins SRp20 and SRp40, resulting in abnormal hTERT splicing. CD4^+^ T cells treated with this hTERT-targeting ASO have limited proliferative capacity due to the inhibition of telomerase function. Such antitelomerase splice-switching oligonucleotide-based strategies might help treat lymphoproliferative disorders. Another study involving a non-obese diabetic mice model showed that using ASOs to modify CTLA-4 isoform ratios can significantly change the pathogenic outcomes of type 1 diabetes in mice. Our group has also shown that antisense splice-switching oligonucleotides can effectively regulate the splicing of IL7R exon 6 in CD4^+^ T cells [74]. The study showed that the ASO (anti-sIL7R) designed to promote exon 6 inclusion reduces the sIL7R secretion from CD4^+^ T cells (Figure 2). It also corrects for the enhanced exclusion of IL7R exon 6 in minigene reporter constructs carrying the MS-risk allele, which increases IL7R exon 6 skipping. Such targeted approaches for correcting splicing defects in T cell factors have the potential to be widely applicable in the treatment of autoimmune diseases, cancers, and viral infections.

## 7. Conclusions

T cells play a crucial role in adaptive immunity by protecting the body from harmful invaders such as pathogens, allergens, and cancer. The process of RNA alternative splicing plays a crucial role in the regulation of T cell development and function. Our search of the ClinVar database supports this, as we found several splice site mutations in T cell factors linked to autoimmune disorders. However, this may not fully represent the effect of RNA splicing in T cells, as it does not consider other *cis* and *trans*-acting mutations. Developing a thorough understanding of the mechanisms that regulate alternative splicing changes in T cell factors will help us design targeted approaches such as splice-switching antisense oligonucleotides to regulate T cell function. Splice-switching antisense oligonucleotides might be another valuable addition to the existing T-cell-based immunotherapeutic approaches, such as chimeric antigen receptor (CAR) T therapy. 

## Figures and Tables

**Figure 1 genes-14-01896-f001:**
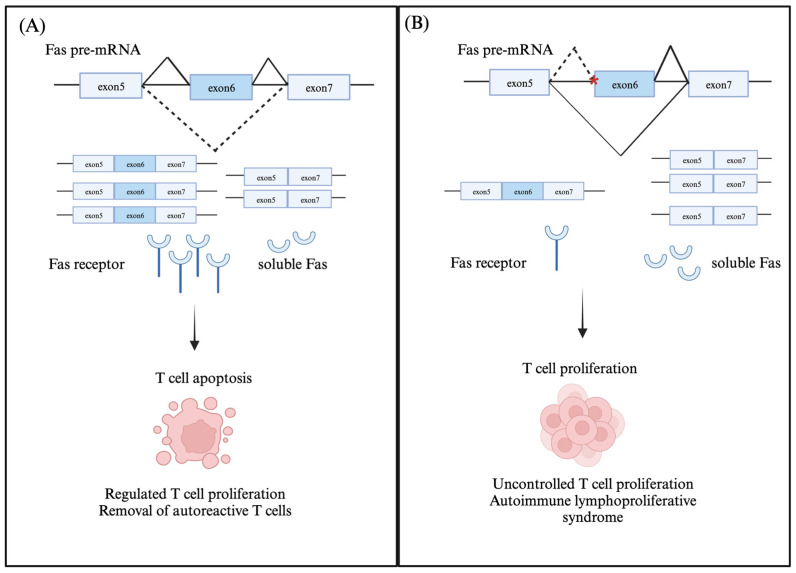
Alternative splicing of Fas exon 6 in activated T cells. (**A**) Fas exon 6 is alternatively spliced to produce two transcript isoforms: the exon 6—included isoform encoding for membrane-bound Fas receptor and the exon 6—skipped isoform encoding for soluble Fas protein. Activated T cells express higher levels of the membrane-bound Fas receptor, which is required for apoptotic signaling in these cells, and thus inhibit uncontrolled T cell proliferation. (**B**) The presence of an SNP in the 3′SS of intron 5 (indicated in red) in the *Fas* gene results in an increased skipping of exon 6 and skews the isoform ratios of Fas in activated T cells. Increased levels of soluble Fas isoform results in increased and unregulated T cell proliferation.

**Figure 2 genes-14-01896-f002:**
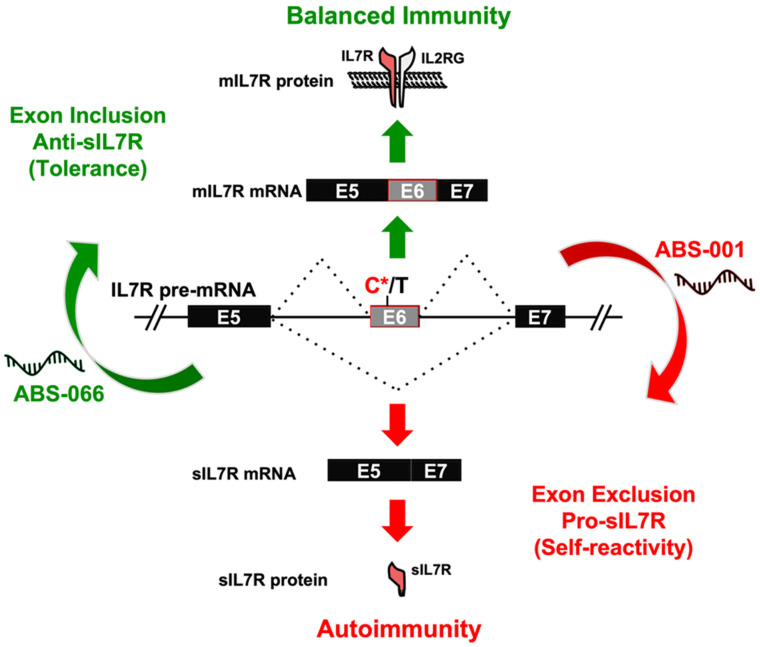
**Antisense modulation of IL7R splicing controls sIL7R expression.** Expression of the membrane-bound (mIL7R) and soluble (sIL7R) isoforms of IL7R is determined by alternative splicing of IL7R exon 6, and this splicing event is influenced by the MS risk SNP rs6897932 (C*/T, where C* is the risk allele). The risk C* allele of rs6897932 enhances the exclusion of exon 6 and the levels of sIL7R, leading to the development of autoimmunity. The expression of sIL7R can be controlled for therapeutic intervention by modulating the splicing of exon 6 with antisense oligonucleotides (ASOs): ABS-066 promotes the inclusion of exon 6 and reduces sIL7R for the treatment of autoimmunity, whereas ABS-001 promotes the exclusion of exon 6 and enhances sIL7R to enhance antitumor immunity.

**Table 1 genes-14-01896-t001:** List of splice site mutations in T cell factors associated with autoimmunity.

Gene	Gene Description	Splice Site Mutation	Genomic Region	Condition(s)
CTLA4	cytotoxic T-lymphocyte-associated protein 4	NM_005214.5(CTLA4):c.458-1G>T	Intron 2	Autoimmune lymphoproliferative syndrome due to CTLA4 haploinsuffiency
		NM_005214.5(CTLA4):c.458-1G>C	Intron 2	
FAS	Fas cell surface death receptor	NM_000043.6(FAS):c.506-1G>C	Intron 5	Autoimmune lymphoproliferative syndrome type 1
		NM_000043.6(FAS):c.334+2T>C	Intron 3	
		NM_000043.6(FAS):c.334+2dup	Intron 3	
		NM_000043.6(FAS):c.335-2A>G	Intron 3	
		NM_000043.6(FAS):c.569-2A>C	Intron 6	
		NM_000043.6(FAS):c.651+1G>A	Intron 7	
		NM_000043.6(FAS):c.651+1G>T	Intron 7	
		NM_000043.6(FAS):c.651+2T>A	Intron 7	
		NM_000043.6(FAS):c.651+2T>C	Intron 7	
		NM_000043.6(FAS):c.651+2_651+3insTGAAAT	Intron 7	
		NM_000043.6(FAS):c.652-1G>A	Intron 7	
		NM_000043.6(FAS):c.676+1G>A	Intron 8	
		NM_000043.6(FAS):c.676+1G>C	Intron 8	
		NM_000043.6(FAS):c.676+1G>T	Intron 8	
FOXP3	forkhead box P3	NM_014009.4(FOXP3):c.736-2A>T	Intron 7	Insulin-dependent diabetes mellitus, Secretory diarrhea syndrome
		NM_014009.4(FOXP3):c.-23+1G>T	Exon1-intron1	
		NM_014009.4(FOXP3):c.210+1G>A	Intron 2	
		NM_014009.4(FOXP3):c.210+1G>T	Intron 2	
IL7R	interleukin 7 receptor	NM_002185.5(IL7R):c.221+2T>G	Intron 2	Immunodeficiency 104, Severe combined immunodeficiency syndrome
		NM_002185.5(IL7R):c.221+1G>A	Intron 2	
		NM_002185.5(IL7R):c.537+1G>A	Intron 4	
		NM_002185.5(IL7R):c.[134A>C;537+1G>A]	Intron 4	
		NM_002185.5(IL7R):c.707-2A>G	Intron 5	
		NM_002185.5(IL7R):c.83-1G>A	Intron 1	
		NM_002185.5(IL7R):c.221+2T>G	Intron 2	
		NM_002185.5(IL7R):c.538-1G>A	Intron 4	
LCK	LCK proto-oncogene, Src family tyrosine kinase	NM_005356.5(LCK):c.481+2T>G	Intron 6	Severe combined immunodeficiency due to LCK deficiency
ZAP70	zeta chain of T-cell-receptor-associated protein kinase 70	NM_001079.4(ZAP70):c.703-1G>A	Intron 5	ZAP70-related severe combined immunodeficiency
		NM_001079.4(ZAP70):c.791-1G>A	Intron 6	

Table 1: A list of all the splice site mutations in T cell factors as discussed in the review. These splice site mutations are classified as pathogenic in the ClinVar database as they are associated with immune disorders. The table also indicates the diseases associated with the splice site mutations in the indicated gene.

## Data Availability

All data are presented in the manuscript and are freely available.

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
