# Peer review of "Role of RNA Alternative Splicing in T Cell Function and Disease"

_genes, 2023, doi:10.3390/genes14101896_

Round 1

Reviewer 1 Report

This review offers a comprehensive study on RNA alternative splicing forms and their functions in T cells. The author effectively covers cell surface reporters, intercellular signaling factors, and transcription factors, providing examples to clarify their roles in T cell function. However, there are a few suggestions and one notable omission that could enhance the review:

1. Include Apoptosis Gene Splicing: It would be beneficial to incorporate information on alternative splicing of apoptosis genes and how it influences T cell function, as highlighted in a recent study (Davia Blake et al., eLife 11:e80953, 2022).

 Minor Suggestions:

1. Introduction Section: Consider adding a figure that illustrates the RNA splicing mechanism. This visual aid can assist readers in understanding the entire process more clearly.

2. Discussion Part: In the discussion, it would be valuable to explore the potential applicability of small-molecule modulators that have seen wide usage in cancer, such as RBM39 degraders and phenothiazine derivatives targeting U2AF–RNA interactions, to T cell treatment. Additionally, discussing any limitations or challenges associated with using these small molecules in the context of T cell therapy would be informative.

In conclusion, this review offers valuable insights into T cell RNA alternative splicing and its potential therapeutic applications. By incorporating the suggested changes and addressing the omission regarding apoptosis gene splicing, this review could become even more comprehensive and impactful. Overall, I appreciate the depth of thought this review provides, and it offers a unique perspective on the topic.

Author Response

  1. Include Apoptosis Gene Splicing: It would be beneficial to incorporate information on alternative splicing of apoptosis genes and how it influences T cell function, as highlighted in a recent study (Davia Blake et al., eLife 11:e80953, 2022).

           Author’s response: This paper has been referenced in the introduction                   and in the section that discusses factors responsible for global changes in             alternative splicing in T cells. They have been highlighted in red in the                      manuscript.

 Minor Suggestions:

  1. Introduction Section: Consider adding a figure that illustrates the RNA splicing mechanism. This visual aid can assist readers in understanding the entire process more clearly.

          Author’s response: We appreciate this suggestion, and we have included a            new figure (now Figure 1) depicting the same.

  1. Discussion Part: In the discussion, it would be valuable to explore the potential applicability of small-molecule modulators that have seen wide usage in cancer, such as RBM39 degraders and phenothiazine derivatives targeting U2AF–RNA interactions, to T cell treatment. Additionally, discussing any limitations or challenges associated with using these small molecules in the context of T cell therapy would be informative.

         Author’s response: We thank the reviewer for this suggestion, and we                     added a short description on how RBM39 modulators may have                               therapeutic potential in correcting alternative splicing changes in T cells.

In conclusion, this review offers valuable insights into T cell RNA alternative splicing and its potential therapeutic applications. By incorporating the suggested changes and addressing the omission regarding apoptosis gene splicing, this review could become even more comprehensive and impactful. Overall, I appreciate the depth of thought this review provides, and it offers a unique perspective on the topic.

Reviewer 2 Report

Summary: there have been lots of reviews published on the importance of alternative splicing in disease processes. However this particular review is focused on the importance of alternative splicing specifically in T cell function and disease, and to my knowledge, it is the first to do so systematically, and will therefore be of interest to colleagues working in that field. The review draws attention to the effect of mutations that disrupt splicing in T cells, and also suggests some potential therapeutic approaches that could be developed centered on the modification of AS in T cells. The review is very well written, well researched and of general interest to the readership of the journal.  To further enhance the manuscript, I would like to suggest the following modifications.

1. The introduction presents, in detail, a series of examples of alternative splicing events that are specific to T cells. I appreciate there are many, but I think the review's message and its overall readability would be enhanced by including a figure / schematic that shows the nature and structure of a key selection of T cell-specific splice isoforms, and their functional significance. This would then synergise with the information presented in Table 1 and throughout the review.

2. The review is very well researched and extremely detailed, which is of course a strength, but it can affect to some extent readability and I notice a number of very long paragraphs. I wonder if the authors could consider whether they could enhance the paragraph structure and overall flow and readability of some of the "fact-heavy" sections.

3. The section on "factors mediating global AS changes in T cells" is of high interest, and I appreciate there is a lot of complexity in terms of describing effectively what might be key in the regulation of AS in T cells. I also feel that this section could benefit from a figure or perhaps a table that lists some of the proposed AS regulators and their effect on the regulation of expression of key T cell-relevant splice isoforms.

4. Also on the subject of regulators of AS, again, I appreciate it's a complex question - I wonder if there might be scope for bringing regulatory ncRNAs into the narrative here; might there be ncRNAs (both miRNAs and perhaps lncRNAs, circRNAs etc) that contribute to the regulation of T cell AS?

Author Response

  1. The introduction presents, in detail, a series of examples of alternative splicing events that are specific to T cells. I appreciate there are many, but I think the review's message and its overall readability would be enhanced by including a figure / schematic that shows the nature and structure of a key selection of T cell-specific splice isoforms, and their functional significance. This would then synergise with the information presented in Table 1 and throughout the review.

Author’s response: We thank the reviewer for this suggestion, and we have included a new figure (now Figure 1) to depict how changes in Fas alternative splicing can alter T cell function.

  1. The review is very well researched and extremely detailed, which is of course a strength, but it can affect to some extent readability and I notice a number of very long paragraphs. I wonder if the authors could consider whether they could enhance the paragraph structure and overall flow and readability of some of the "fact-heavy" sections.

Author’s response: We thank the reviewer for this suggestion, and we have structured the review in distinct sections to cover different concepts.

  1. The section on "factors mediating global AS changes in T cells" is of high interest, and I appreciate there is a lot of complexity in terms of describing effectively what might be key in the regulation of AS in T cells. I also feel that this section could benefit from a figure or perhaps a table that lists some of the proposed AS regulators and their effect on the regulation of expression of key T cell-relevant splice isoforms.

Author’s response: We appreciate this suggestion from the reviewer, and Figure 2 in the review depicts exactly what the reviewer is asking for with the example of ASOs modulating IL7R exon 6 alternative splicing in T cells.

  1. Also on the subject of regulators of AS, again, I appreciate it's a complex question - I wonder if there might be scope for bringing regulatory ncRNAs into the narrative here; might there be ncRNAs (both miRNAs and perhaps lncRNAs, circRNAs etc) that contribute to the regulation of T cell AS?

Author’s response: We appreciate this suggestion and we have added a short description on how lncRNA can modulate Fas pre-mRNA splicing in the section that discusses Fas alternative splicing. This has been highlighted in red.